# The Impact of Sleep-Disordered Breathing on Ghrelin, Obestatin, and Leptin Profiles in Patients with Obesity or Overweight

**DOI:** 10.3390/jcm11072032

**Published:** 2022-04-05

**Authors:** Piotr Pardak, Rafał Filip, Jarosław Woliński

**Affiliations:** 1IBD Unit, Department of Gastroenterology, Kliniczny Szpital Wojewódzki Nr 2 im. Św. Jadwigi Królowej w Rzeszowie, Medical College of Rzeszów University, 35-301 Rzeszów, Poland; r.s.filip@wp.pl; 2Department of Internal Medicine, Medical College of Rzeszów University, University of Rzeszow, 35-310 Rzeszow, Poland; 3Department of Internal Medicine, Institute of Rural Health, 20-090 Lublin, Poland; 4Department of Animal Physiology, The Kielanowski Institute of Animal Physiology & Nutrition, Polish Academy of Sciences, 05-110 Jabłonna, Poland; jarek.wolinski@gmail.com

**Keywords:** obesity, sleep-disordered breathing, ghrelin, leptin, obestatin, sleep apnea

## Abstract

Background: The impact of concomitant obesity and sleep disorders on neuropeptides related to energy balance is poorly understood. The aim of this study was to assess the nocturnal profile of total ghrelin, obestatin, and leptin in patients with elevated BMI and to investigate the impact of breathing-related sleep disorders on these hormone levels. Methods: The study involved 58 patients with suspicion of obstructive sleep apnea (OSA). Patients underwent anthropometric and sleep examination and measurements of night ghrelin, leptin, and obestatin levels. Results: In patients with OSA (*n* = 46), recognized on the basis of sleep examination outcomes, the correlation of anthropometric measurements with parameters of sleep disorders and ghrelin levels was observed, contrary to the control group (*n* = 12). In the OSA group, levels of ghrelin were significantly lower than in the control group at 5:00 and 7:00. Levels of leptin in the OSA group were also lower than those in the control groups (not statistically significant). Profiles of obestatin in both groups were similar. Conclusions: Our results confirm the relationship between obesity and sleep-disordered breathing. Both these disorders affect ghrelin levels—parameters of obesity negatively correlate with hormone concentration, and OSA seems to lower ghrelin values in the second half of the night.

## 1. Introduction

Obesity, defined as the excessive or abnormal accumulation of fat tissue in the body, is diagnosed based on a body mass index (BMI) value of at least 30 kg/m^2^. This disease affects approximately 13% of the world’s adult population, and over the past 40 years, the number of obese people has tripled. Obesity adversely affects all aspects of human health and increases the risk of developing many diseases, including type 2 diabetes, cardiovascular disease, metabolic syndrome, and various cancers [1,2].

The primary cause of obesity is an energy imbalance between calories consumed and calories expended resulting from increased consumption of high-energy foods and low physical activity. However, genetic/familial factors or hormonal disorders also play an important role in the development of obesity [2,3].

Population-based studies indicated that the risk of obesity is also significantly increased by sleep deprivation. Lifestyle changes made sleep deficiency commonplace in recent decades. Increasing working hours, stress, work-related rush, widespread access to social media, and depressive disorders, particularly exacerbated in the era of the COVID-19 epidemic, adversely affect the quantity and the quality of sleep [4]. Sleep deprivation can promote obesity in two ways: on the one hand, it leads to hedonic eating and more frequent high-calorie meal choices, and on the other, it causes sleepiness during the day, which impedes physical activity [5,6,7].

In turn, excessive weight is a major risk factor for obstructive sleep apnea (OSA) because fatty tissue accumulated in the neck region narrows the airway reducing the airflow and also contributes to hypotonia of the pharyngeal muscles and repeated closure of the upper airway at the level of the throat during sleep, leading to apnea and multiple awakenings.

Although sleep duration is normal, a decrease in sleep efficiency occurs, and as a result, patients experience daytime symptoms such as drowsiness, fatigue, impaired concentration, and metabolic disturbances. Further consequences include stimulation of the sympathetic nervous system, increased inflammatory parameters, an increased risk of developing insulin resistance, hyperinsulinemia, and glycemic disturbances with a tendency toward hyperglycemia and abnormal glucose tolerance [8,9]. The prevalence of OSA ranges widely from 6% to as much as 49%, depending on the studied population and diagnostic criteria; however, it is thought that these data are under-reported [10].

Obesity has been recognized as one of the main risk factors for OSA. According to epidemiological studies, up to 60% of OSA cases are the result of obesity. In the adult population, the prevalence of OSA is estimated to be above 50% in obese subjects. Numerous studies correlated weight gain and loss with increasing and decreasing the severity of OSA, respectively. Therefore, the primary treatment for OSA in obese patients is weight reduction. The effectiveness is confirmed by studies in which a reduction in the severity of OSA and an improvement in the quality of night sleep were observed in response to weight loss caused by surgery or diet [11,12,13].

Obesity and sleep disorders are thus linked by a bidirectional causal relationship, and identifying effective methods to interrupt this mechanism is important for patients’ prognosis and their quality of life. Numerous mechanisms are involved in the regulation of energy balance, one of which is the action of the neuropeptides that regulate energy homeostasis: ghrelin, leptin, and obestatin. 

Ghrelin is synthesized mainly by the cells of stomach gastric mucosa, but it is also found in the hypothalamus, pituitary, and intestines. This hormone has orexigenic action, i.e., it stimulates food intake, so under physiological conditions, its concentration in blood increases during starvation and decreases after eating [14,15]. Paradoxically, in patients with obesity, fasting ghrelin concentrations are lower than in subjects with normal weight. However, after eating, reductions in ghrelin concentrations occur to a small extent in obese individuals, leading to the persistence of relatively high postprandial values and excessive food intake [16,17].

On the contrary, the conclusions from the studies of the influence of OSA on ghrelin levels differ. In the only study evaluating the effect of ghrelin profile, no differences were found depending on the diagnosis of OSA [18]. Most of the studies mainly involved morning measurements, and their results were dominated by observations showing higher values or no effect of OSA on ghrelin levels [19,20,21,22,23]. The levels of ghrelin in the course of OSA were only lower in one study [24].

Leptin is secreted by adipocytes, and its concentration in blood correlates with the amount of adipose tissue, which is an indicator of body energy status. Under physiological conditions, leptin reduces hunger and enhances metabolism; hence, its concentration decreases during starvation and increases during eating [15,25]. In obesity, the leptin level is elevated due to increased adipocytes and also due to leptin resistance, which is characterized by a lack of appetite suppression despite high leptin concentrations [26].

In the majority of reports, diagnosis of OSA correlated with higher leptin concentrations; less frequently, no deviations in leptin levels were found. We did not find any reports where OSA correlated with lower leptin levels. A review of the literature by Mashaqi et al. indicated greater importance of obesity and adipose tissue on leptin levels than OSA [19]. However, few studies show a decrease in leptin levels in response to CPAP therapy [19,27,28]

Obestatin is the least understood of the mentioned neuropeptides, and its importance in metabolism is still under investigation. What is known is that obestatin decreases nutrient absorption and intestinal peristalsis. Its concentration is lower in individuals with obesity than in those with normal weight, and its diurnal fluctuations are much smaller than those of ghrelin [29,30]. Obestatin is thought to play a role in the regulation of glucose and fat metabolism and the reduction in inflammation and to have a beneficial effect on the preservation of organ function by stimulation of proliferation and inhibition of apoptosis of cells [31].

The available studies showed no significant deviations of obestatin in OSA. In a study by Liu et al., in OSA, obestatin values were lower, but this difference did not reach the level of statistical significance [24]. While in the study by Zirlik et al., obestatin levels between OSA patients and the control group were similar; moreover, CPAP therapy did not affect the level of obestatin [27].

The relationship between the values of the mentioned neuropeptides related to energy balance and obesity is relatively well understood. Obesity, in turn, remains in a reciprocal relationship with sleep disorders.

Therefore, the aim of this observational study was to assess the nocturnal profile of total ghrelin, obestatin, and leptin in patients with elevated BMI and investigate the impact of breathing-related sleep disorders on the levels of appetite-regulating hormones.

## 2. Materials and Methods

### 2.1. Study Group

The study was conducted on patients with excessive weight (BMI ≥ 25 kg/m^2^) hospitalized in the Department of Internal Medicine of the Institute of Rural Medicine in Lublin due to clinical suspicion of sleep-disordered breathing. The suspicion was based on the presence of symptoms such as pauses in breathing during sleep, snoring, or excessive daytime sleepiness. Patients with decompensation of chronic cardiovascular or respiratory diseases and those taking medications with potential effects on sleep parameters were excluded from the study, as were patients with factors that might affect ghrelin, leptin, or obestatin levels, such as significant gastrointestinal pathology (active inflammatory bowel disease, celiac disease) or a history of abdominal surgery (e.g., gastrectomy, intestinal resection).

A total of 58 subjects were eligible to participate in the study, and all of them gave informed consent. The study was approved by the Bioethics Committee of the Institute of Rural Medicine (Decision No. 6/2014).

The population described in this article served as a study group in our other research, the results of which were published in the article “Associations of Obstructive Sleep Apnea, Obestatin, Leptin, and Ghrelin with Gastroesophageal Reflux” [32].

### 2.2. Anthropometric Data

All participants underwent a physical examination. The BMI was calculated as the body weight in kilograms divided by the height in meters squared (kg/m^2^). Overweight was diagnosed if BMI was in the range of 25–29.9 kg/m^2^, and obesity was recognized with BMI ≥ 30 kg/m^2^ (30–34.9 kg/m^2^—class I, 35–39.9 kg/m^2^—class II, and > 40 kg/m^2^—class III). The waist circumference (WC), i.e., the smallest circumference between the rib cage and the iliac crest, was measured in the standing position.

### 2.3. Sleep Examination

An all-night sleep examination was performed according to the current guidelines of the American Academy of Sleep Medicine (AASM) [33]. Sleep assessment was carried out in all patients based on polygraphic testing (type III according to AASM) using an Embletta MPR PG (formerly Embla, currently Natus; Pleasanton, CA, USA). Additionally, in 12 patients (10 from the OSA group and 2 from the control group), polysomnography was performed (type II according to AASM) with EmblaS4500 devices (formerly Embla, currently Natus; Pleasanton, CA, USA) and RemLogic diagnostic software (formerly Embla, currently Natus; Pleasanton, CA, USA). Categorization of respiratory incidents and assessment of sleep apnea severity was performed according to standard criteria by an experienced physician. The following parameters of sleep disorders assessed by polysomnography were selected for statistical analyses: apnea/hypopnea index (AHI), indicating the number of apneas/hypopneas occurring per hour of sleep; percentage snoring time during sleep; and the value of the mean and lowest saturation during sleep (SpO_2_ mean; SpO_2_ lowest). According to current guidelines, OSA was diagnosed when AHI ≥ 15 or AHI ≥ 5 in the presence of clinical symptoms. The disorder was classified as “mild”, “moderate”, or “severe” when the number of episodes per hour was 5–14, 15–29, or > 30, respectively [34].

### 2.4. Laboratory Measurements

On the day after the sleep examination, blood was drawn from a catheter inserted into a peripheral vein at the following times: 23:00, 01:00, 03:00, 05:00, and 07:00. Drawing blood samples did not require waking the patient. The first blood sample was taken at least three hours after the end of supper, and the last blood sample was drawn at least 60 min before breakfast. Patients were instructed to refrain from meals on the night of blood sampling. For one patient, only one sample at 23:00 could be obtained because the patient did not consent to further samples being taken. Blood was collected into tubes containing EDTA and then centrifuged at 3000 rpm for 20 min at 4 °C. The obtained blood serum was stored in Eppendorf tubes at −80 °C until assays were performed. Hormonal assays were performed using radioimmunoassays (RIA): total ghrelin—Ghrelin (Human) RIA Kit (EMD Millipore’s Corp. Inc. Billerica, MA, USA); leptin—Leptin (Human) RIA Kit (EMD Millipore’s Corp. Inc. Billerica, MA, USA); obestatin—Obestatin (Human, Monkey) RIA Kit (Phoenix Pharmaceuticals, Inc. Burlingame, CA, USA). For technical reasons, obestatin and leptin determinations were not performed in all patients (*n* = 39–42).

### 2.5. Statistical Analysis

Statistical analyses were performed using the software package Statistica (data analysis software system), version 13 (TIBCO Software Inc., 2017; Palo Alto, CA, USA).

Descriptive statistics were computed for the analyzed parameters as appropriate: mean with standard deviation (SD) for parameters with normal distribution or median and range for parameters without normal distribution.

Spearman rank correlation analysis was used to analyze the relationship between hormone determinations and parameters of obesity and sleep. A comparison of neuropeptide levels in subgroups with and without OSA was performed with the Mann–Whitney test, and the Kruskal–Wallis test was used in the comparison of neuropeptides levels according to the degree of obesity (based on BMI values). Results for which the *p*-value was below 0.05 were considered statistically significant.

Additionally, a multivariate model for the whole study group was constructed with mean ghrelin concentration as the dependent variable and BMI, AHI, gender, OSA diagnosis, and lowest and mean saturation as independent variables. Linear regression was used, and due to asymmetric distribution, ghrelin values were logarithmized before analysis. All algorithms led to the same univariate model with BMI as the independent variable.

## 3. Results

### 3.1. Characteristics of the Study Population

The study was conducted on 58 subjects (48 males and 10 females) aged between 34 and 75 years (mean = 54.5; SD = 11.2). Based on BMI values, it was noted that most patients (44 subjects, 75.9%) were obese, with class I, II, and III of obesity recognized in 17, 12, and 15 subjects, respectively. In other patients (14 subjects, 24.1%), overweight was diagnosed. The characteristics of the study group are presented in Table 1.

The study group had comorbidities mainly related to the circulatory system and metabolic disorders. The most common were: arterial hypertension (47 patients, 81%); gastroesophageal reflux disease (40 patients, 69%); dyslipidemia (34 patients, 59%), carbohydrate metabolism disorders in 23 patients (40%) of which 14 patients (24%) had type 2 diabetes and 9 patients (16%) had pre-diabetes; coronary heart disease (16 patients, 28%); hyperuricemia (10 patients, 17%); and heart failure (4 patients; 7%). The participants in the study mainly used medications due cardiovascular diseases. These were drugs included in the following groups:: beta blockers (37 patients); diuretics (32 patients), angiotensin-converting enzyme inhibitors (28 patients), and angiotensin receptor blockers (8 patients); calcium channel blockers (18 patients); doxazosin (6 patients); clonidine (3 patients); antiplatelet agent (acetylsalicylic acid and/or clopidogrel in 17 patients); anticoagulant (warfarin or rivaroxaban in 7 patients); statin (22 patients); fibrate (9 patients).

### 3.2. Obstructive Sleep Apnea Assessment

Based on AHI values, OSA was diagnosed in 46 patients (79.3%)—severe in 31, moderate in 12, and mild in three patients. These patients were included in the OSA group. The remaining 12 patients served as the control group.

The anthropometric parameters did not differ significantly between subgroups except for weight. However, when only men were analyzed, the BMI and weight values were significantly higher in the OSA group than in the control group.

Among the evaluated polygraphic parameters, only SpO_2_ lowest significantly differed between the OSA and control subgroups. The characteristics of the subgroups are presented in Table 1.

### 3.3. Neuropeptide Profile Evaluation

In the whole study group of patients with elevated BMI, similar patterns of nocturnal blood levels were observed for ghrelin and leptin, with median, minimum values at 01:00. The night profile for obestatin had almost the opposite pattern. The results of the neuropeptide determinations are presented in Figure 1, Figure 2 and Figure 3.

### 3.4. Association of Anthropometric Measurements with Sleep Disorder Parameters

The group of patients with OSA comprised 36 (78%) patients with obesity and 10 (22%) patients with overweight. Evaluating the occurrence of OSA in the context of BMI values, it was noted that the highest proportion of OSA was noted in patients with BMI > 40 kg/m^2^ (Table 2).

A statistically significant correlation was observed between anthropometric measurements (BMI and WC) and sleep disorder parameters (except for AHI) in the whole study population, and in the OSA group, no significant correlations were noted in the control group (Table 3).

### 3.5. Association of Gender and Anthropometric Measurements with Neuropeptide Levels

Among the evaluated neuropeptides, only ghrelin levels correlated with obesity parameters. In the OSA group, significant negative correlation was found between BMI and average median ghrelin level (−0.32, *p* = 0.03) and ghrelin levels at 23:00 (−0.41, *p* = 0.005) and at 01:00 (−0.40, *p* = 0.007), and also between WC and ghrelin levels at 23:00 (−0.36, *p* = 0.01) and at 01:00 (−0.37, *p* = 0.01). No significant correlation between these parameters was observed in the control group.

In subgroups of patients classified according to BMI, average median values of ghrelin and the majority of night ghrelin levels were highest in patients with overweight and lowest in patients with class II and III obesity (Figure 4). Statistically significant differences were noted between ghrelin values at 23:00 (*p* = 0.002), at 01:00 (*p* = 0.02), and at 03:00 (*p* = 0.04) and between average median values of ghrelin (*p* = 0.01).

No correlation between leptin or obestatin levels and anthropometric measurements was observed in the OSA group or in the control group.

The levels of neuropeptides were compared between males and females. Males had significantly lower levels of ghrelin than females (Me = 411; Range: 212–1928 vs. Me = 499; range: 407–861). However, after including gender in the regression analysis, it turned out to be an insignificant variable. The observed difference probably results from a significantly lower value of BMI and weight in the group of women participating in the study. No significant differences in the levels of obestatin or leptin depending on gender were found.

### 3.6. Association of Sleep Disorder Parameters with Neuropeptide Levels

The results of neuropeptide determinations were compared between the group with OSA and the control group in the context of the average median value (Table 4) and median values within the night profile. In the OSA group, the levels of ghrelin were lower than in the control group. However, the differences were statistically significant only for values at 05:00 and 07:00 (Figure 5). Levels of leptin in the OSA group were also lower than those in the control group (not statistically significant), but the profiles had a similar pattern (Figure 6). Profiles of obestatin in the OSA and in the control group were similar in pattern and in values (Figure 7).

In the next step, the correlation between neuropeptide levels and sleep parameters (AHI, snoring time, SpO_2_ mean, SpO_2_ lowest) was assessed. However, no significant correlation was found except for a weak correlation, near statistical significance, between the lowest saturation value during sleep and ghrelin levels at 23:00 and 01:00.

### 3.7. Multivariate Analysis of the Association between BMI, Sleep Disorder Parameters, and Ghrelin Levels

Multivariate analyses confirmed an association between ghrelin concentrations and BMI, and a lack of association between ghrelin levels and OSA diagnosis, gender, sleep time, and sleep disturbance parameters. The value of the coefficient of determination indicated that almost one-tenth of the variation in ghrelin concentration resulted from BMI changes. The negative value of the regression coefficient demonstrates that the higher the BMI, the lower the ghrelin concentration (Table 5).

## 4. Discussion

The relationship between obesity and sleep disorders was proven in many studies [10]; however, many issues still require explanation. One of them is the influence of OSA combined with obesity on levels of ghrelin, leptin, and obestatin, i.e., neuropeptides strongly related to body energy homeostasis. In order to assess this impact, we performed a study on a group of patients with elevated BMI (≥ 25 kg/m^2^) and OSA and compared results with a control group similar in terms of BMI but without OSA.

First, our outcomes confirmed the relationship between obesity and sleep-related breathing disorders because the majority (82%) of studied patients with obesity had OSA diagnosed. Moreover, OSA was diagnosed most frequently in patients with class III obesity, which is consistent with the assumption that more than half of the persons with obesity (especially III class) are affected by OSA [35,36]. In the studied patients with OSA, values of BMI and WC were higher and significantly correlated with saturation during sleep (negative correlation) and snoring time (positive correlation), in contrast to the control group with lower mean values of anthropometric parameters that did not correlate with sleeping disorder parameters.

Numerous studies addressed this topic, showing that habitual sleep duration below 7.7 h leads to an increase in BMI values and that the parameters of obesity, to a limited extent, allow the identification of patients at high risk of OSA and even predict the severity of breathing disorders during sleep [37,38].

The mechanism underlying the association between sleep and body weight is multifactorial; however, homeostatic control seems to play a key role. It involves increased hunger during sleep deprivation through changes in levels of leptin that suppresses appetite and ghrelin, a peptide that stimulates appetite [39,40,41]. Hence, alterations in these neuropeptide levels promote obesity, while obesity modifies the concentration of neuropeptides.

In people with normal weight, the levels of plasma ghrelin during the day show a pre-prandial increase and postprandial decrease, and at night the concentration increases, reaching the maximum at about 01:00–02:00, followed by a decline until morning awakening with a minimum value at 09:00 [14,42,43]. In turn, in subjects with obesity, ghrelin levels are lower than in normal-weight people, which may be explained by chronic excessive food intake that inhibits ghrelin secretion, or it may be associated with hyperinsulinemia, which is commonly found in obesity. It was proven that patterns of ghrelin and insulin secretion are reciprocal, especially after meals [14,44,45]. Interestingly, this lower level of ghrelin in patients with obesity does not lead to weight reduction. It may be influenced by the fact that in obesity, ghrelin, although decreased, displays blunted meal effects, i.e., the amplitude of pre- and post-prandial levels of ghrelin are low [45]. In obesity. the pattern of ghrelin secretion is also altered—the nocturnal increase is blunted or may not occur [45]. Alteration of the ghrelin night profile was also noted in our study conducted on patients with obesity or overweight. We did not observe a night increase in ghrelin, and we even noted a nadir in the ghrelin level at 01:00. The negative correlation between ghrelin and anthropometric parameters observed in our study confirmed the impact of obesity on ghrelin levels, which is in accordance with other studies [46,47].

Levels of neuropeptides may be affected not only by obesity but also by sleep disorders; however, available data concerning this issue are not consistent. This is probably due to variability in population characteristics and study design. Therefore, the results of a systematic review and meta-analysis are valuable, suggesting that reduction in sleep duration is associated only with ghrelin change, even though sleep deprivation is thought to impact both appetite-regulating hormones. Nevertheless, both qualitative and quantitative sleep disorders were generally associated with increased ghrelin levels [48]. Such results were observed in subjects with normal weight [37,49] but also—less frequently—in populations with elevated BMI [50]. Other studies indicated similar levels of ghrelin among patients with obesity, either with or without OSA [51,52]. In turn, in our research, ghrelin levels in patients with OSA were lower than values in the control group, but the differences reached statistical significance only for values at 05:00 and 07:00. However, the mean BMI values in our studied groups were higher (about 36 kg/m^2^ and 33 kg/m^2^ in the OSA and control groups, respectively) than in the mentioned papers (30 kg/m^2^ and 28 kg/m^2^), and this might have had an impact on the results.

As was mentioned, ghrelin levels correlated inversely with BMI, and the high BMI values in our study group were followed by low ghrelin concentrations. Hence, it may be assumed that the impact of obesity on the ghrelin level dominates over the impact of OSA (probably via insulin levels). This assumption is supported by the results of studies performed by Sánchez-de-la-Torre et al., who suggested that sleep apnea is not a determinant factor in ghrelin levels and that the hormone concentration is associated mainly with obesity. This study was conducted primarily on overweight and obese subjects with BMI close to those in our study (mean BMI 34 kg/m^2^ and 32 kg/m^2^). They also observed that ghrelin levels in patients with OSA and obesity were lower than in obese controls and also significantly lower than in non-obese patients with OSA [18]. This may be explained by the fact that obesity contributes more to insulin resistance and chronic inflammation than OSA, so it decreases ghrelin levels more. Moreover, no correlation between ghrelin level and sleep disorder parameters was found in our study or in the cited study [18]. However, we noticed that in the OSA group, early morning values of ghrelin were lower than in the control group, and this prompted us to suppose that OSA may enhance the mechanism of ghrelin reduction related to obesity, and therefore it may contribute to overeating. This is because low levels of ghrelin, as well as blunted postprandial decline in ghrelin levels, lead to weaker stimulation of the reward or satiety centers in the brain, which is associated with binge-eating disorder [53,54]. Hence, the question arises of whether OSA treatment improves ghrelin levels and eating habits. The literature reports on the subject are inconclusive [19], but our results may provide a rationale for greater emphasis on the diagnosis and treatment of sleep disorders or the use of new drug classes (e.g., ghrelin antagonists/analogs) in the treatment of obesity.

Although the level of leptin is known to correlate with body weight, and to depend mainly on the amount of adipose tissue [25], in our study, no correlation between obesity parameters and leptin concentrations was found. It might be because we assessed BMI and WC, not body fat. In people with obesity, leptin values are generally higher (mainly due to increased adipocytes and leptin resistance), and the profile shows a peak at 02:00 [45]. We observed neither night elevation of leptin nor differences between leptin levels in groups of patients with and without OSA. 

The available studies indicate that OSA either increases or has no effect on the values of leptin. In the previously quoted article by Mashaqi et al., the authors indicate greater importance of obesity and the amount of adipose tissue than the influence of OSA on leptin levels [19]. Similar conclusions were obtained in a comprehensive study by Arnardotir et al., which also found no deviations in the level of leptin in OSA [55]. In several other studies, higher values of leptin in OSA were observed [27,28]. This is explained by the theoretical consideration that hypoxia-related to OSA causes an increase in leptin, which in turn is supposed to enhance the respiratory response preventing hypercapnia and increasing neural compensatory mechanisms, minimizing upper airway collapse [56,57]. Our results do not support this theory, as no correlation between sleep parameters and leptin levels was found.

Obestatin is the least understood hormone of those analyzed in the study, and research is still being conducted to determine its significance in the body. Previous studies indicated lower obestatin levels in obese subjects than in lean controls [58]. In our study, no association between obestatin and anthropometric parameters was noted, but the study group was rather homogeneous in terms of BMI, and this fact may have influenced the lack of correlation. We also did not observe any association between obestatin levels and OSA, which is in accordance with the observations of other authors [27].

In summary, it can be said that our results confirm the relationship between obesity and sleep-disordered breathing. Both these disorders have an impact on ghrelin levels—parameters of obesity negatively correlate with hormone concentration, and OSA seems to lower ghrelin values in the second half of the night. Therefore, OSA potentially exacerbates ghrelin disturbances associated with obesity and worsens eating disorders. No correlation was observed between leptin and obestatin levels and obesity parameters or OSA. However, further studies in this area are indicated.

## Figures and Tables

**Figure 1 jcm-11-02032-f001:**
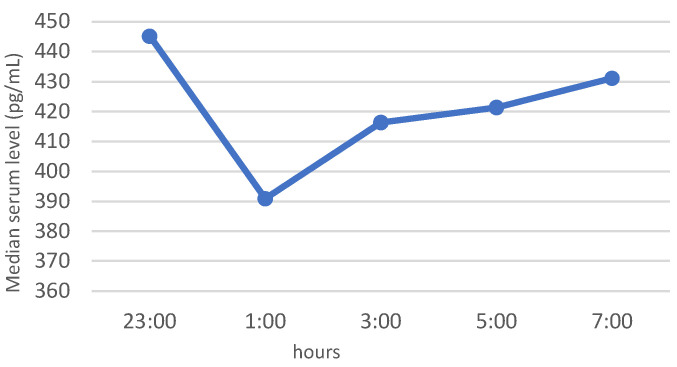
Night profile of ghrelin in the study group.

**Figure 2 jcm-11-02032-f002:**
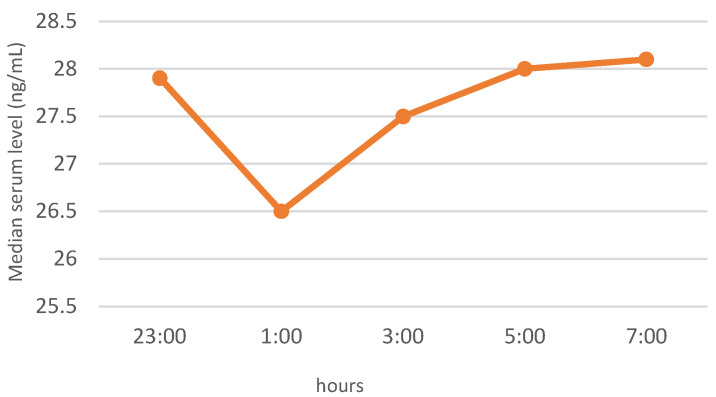
Night profile of leptin in the study group.

**Figure 3 jcm-11-02032-f003:**
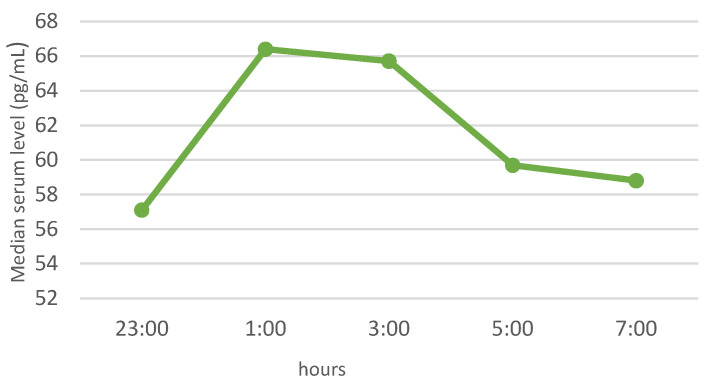
Night profile of obestatin in the study group.

**Figure 4 jcm-11-02032-f004:**
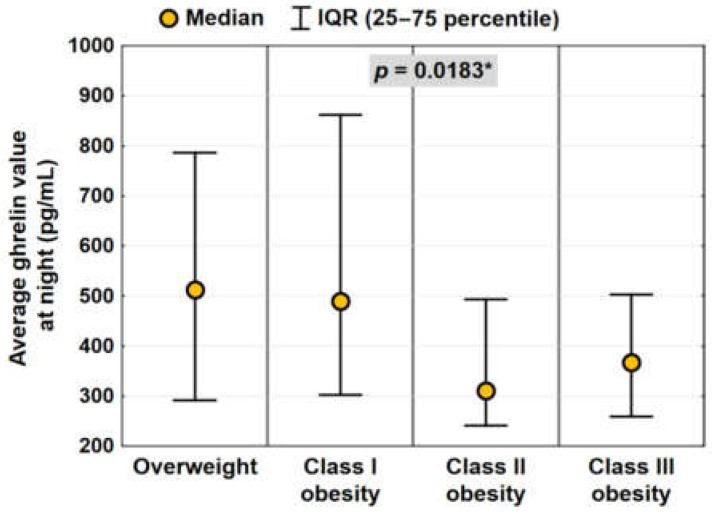
Night levels of ghrelin in subgroups classified according to BMI values (statistically significant correlation is marked using *). IQR—interquartile range.

**Figure 5 jcm-11-02032-f005:**
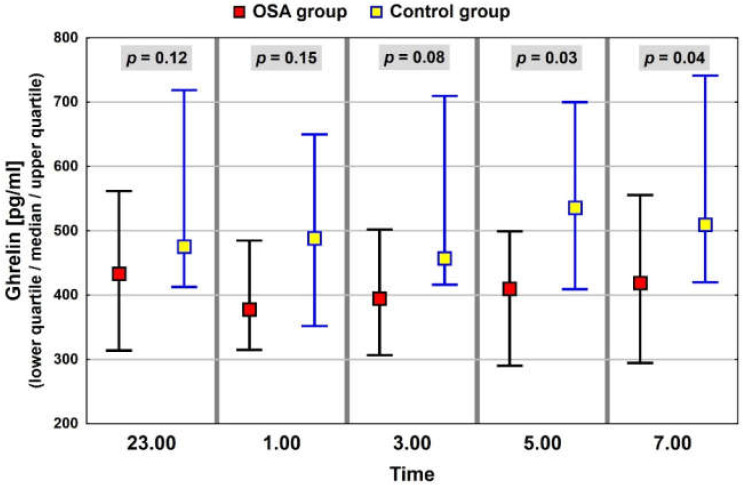
Night ghrelin values in groups of patients with and without OSA. The interquartile range (IQR) of neuropeptide concentrations is marked with black and blue lines.

**Figure 6 jcm-11-02032-f006:**
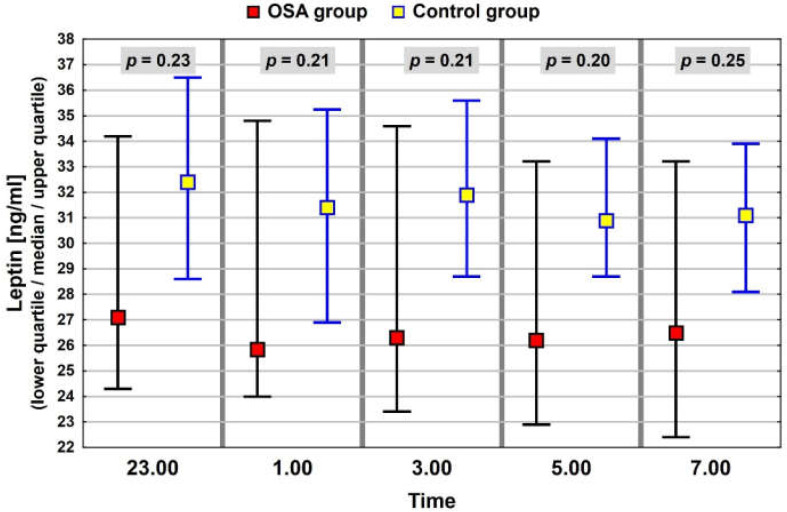
Night leptin values in groups of patients with and without OSA. The interquartile range (IQR) of neuropeptide concentrations is marked with black and blue lines.

**Figure 7 jcm-11-02032-f007:**
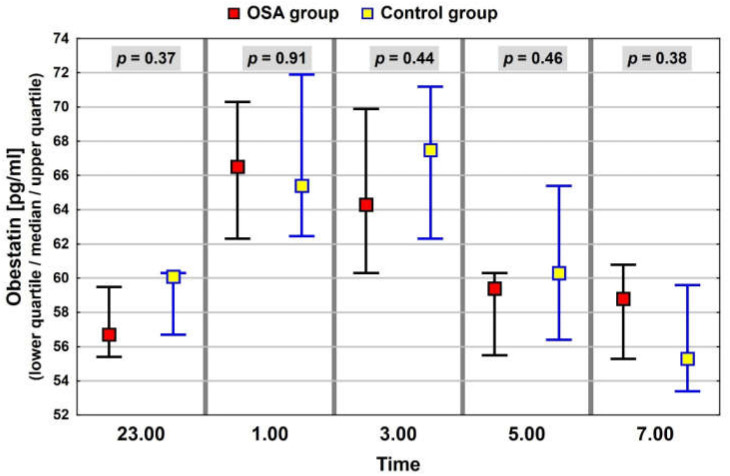
Night obestatin values in groups of patients with and without OSA. The interquartile range (IQR) of neuropeptide concentrations is marked with black and blue lines.

**Table 1 jcm-11-02032-t001:** Basic characteristics of the study group and subgroups, and comparison of polygraphic parameters between the OSA and control groups.

Items	Whole study Population (*n* = 58)		OSA Group (*n* = 46)		Control Group (*n* = 12)	
	mean ± SD	min–max	mean ± SD	min–max	mean ± SD	min–max
Gender [M/F]	48/10		41/5		7/5	
Age (years)	55 ± 11.2	34–75	55 ± 10.6	34–75	57 ± 8.7	46–72
Weight (kg)	105 ± 20.8	77–161	108 ± 21.1	78–161	93 ± 15.0	77–117
BMI (kg/m^2^)	35 ± 6.4	25–50	36 ± 6.5	27–50	33 ± 5.5	25–43
WC (cm)	116 ± 12.8	96–147	117 ± 13.4	96–147	111 ± 9.0	99–122
AHI	38 ± 28.1	0.7–119.9	46 ± 25.8	13.2–119.9	6 ± 4.2	0.7–12.4
Snoring time (%)	14 ± 15.9	0–71	15 ± 16.3	0–71	8 ± 13.1	0–47
SpO_2_ mean (%)	91 ± 4.7	66–97	91 ± 5.2	66–97	93 ± 1.7	91–96
SpO_2_ lowest (%) *****	76 ± 11.9	49–91	74 ± 12.1	49–91	85 ± 4.9	72–90
Sleep time (min.)	449 ± 80	223–557	447 ± 84	223–557	456 ± 65	328–528
Sleep test time (min.)	484 ± 60	266–580	484 ± 63	266–580	485 ± 47	418–543

*** *p* < 0.001; AHI—apnea/hypopnea index; BMI—body mass index; F—female; M—male; OSA—obstructive sleep apnea; SpO_2_ mean—value of mean saturation during sleep; SpO_2_ lowest—value of lowest saturation during sleep; WC—waist circumference; SD—standard deviation.

**Table 2 jcm-11-02032-t002:** Occurrence of obstructive sleep apnea in patients with obesity or overweight.

	Overweight *n* = 14 (*n*,%)	Obesity			
		Class I *n* = 17 (*n*,%)	Class II *n* = 12 (*n*,%)	Class III *n* = 15 (*n*,%)	Total *n* = 44 (*n*,%)
OSA group *n* = 46	10 (71%)	13 (76%)	9 (75%)	14 (93%)	36 (82%)
Control group *n* = 12	4 (29%)	4 (24%)	3 (25%)	1 (7%)	8 (18%)

OSA—obstructive sleep apnea.

**Table 3 jcm-11-02032-t003:** Correlation between parameters of sleep disorders and anthropometric measurements in patients with and without obstructive sleep apnea.

	OSA Group		Control Group	
	BMI	WC	BMI	WC
AHI	0.21	0.19	−0.56	−0.04
Snoring time (%)	0.31 **	0.30 **	−0.13	0.30
SpO_2_ mean (%)	−0.52 **	−0.53 **	−0.57	0.07
SpO_2_ lowest (%)	−0.41 **	−0.39 **	−0.04	0.24

** *p* < 0.05; BMI—body mass index; OSA—obstructive sleep apnea; SpO_2_ mean—value of mean saturation during sleep; SpO_2_ lowest—value of lowest saturation during sleep; WC—waist circumference.

**Table 4 jcm-11-02032-t004:** Average median values (and ranges) of appetite-regulating hormones in patients with and without obstructive sleep apnea. OSA—obstructive sleep apnea.

	OSA Group	Control Group
Ghrelin (pg/mL) *p* = 0.05	426.6 (211.5–1928.0)	496.2 (324.2–1147.1)
Leptin (ng/mL) *p* = 0.19	25.9 (16.5–44.8)	31.9 (21.5–37.1)
Obestatin (pg/mL) *p* = 0.7	61.9 (55.5–98.2)	61.6 (53.9–71.4)

**Table 5 jcm-11-02032-t005:** Results of regression analysis.

Independent Variable	Logarithm of the Mean Ghrelin Concentration R^2^ = 9.7%, F = 6.0, *p* = 0.0175 *		
	B (95% CI)	*p*	β
BMI	−0.020 (−0.037, −0.004)	0.0175 *	−0.31

R^2^—coefficient of determination, BMI—body mass index, F—test statistic and *p*-value for significance of whole model, B—regression coefficient with 95% CI, *p*-value for significance of each regression coefficient, β—standardized regression coefficient (statistically significant correlation is marked using *).

## Data Availability

The data presented in this study are available on request from the corresponding author.

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
