# Peer review of "The Impact of Sleep-Disordered Breathing on Ghrelin, Obestatin, and Leptin Profiles in Patients with Obesity or Overweight"

_jcm, 2022, doi:10.3390/jcm11072032_

Round 1
Reviewer 1 Report
Recommendations for Authors
・Does the introduction provide sufficient background and include all relevant references?
I recommend that you explain a little more detail on the interrelationship between obesity and sleep disorders, and briefly summarize the rest.
・Are the methods adequately described?
It is desirable to add to Table 1 about sleep time, sleep test time and number of people according to type3 and type2, comorbidity and medication.
・Are the results clearly presented?
1) Please correct the item of 3.6 because it is duplicated.
2) Although you examined it separately for males and females only in 3.2, was there any difference between males and females in the neuropeptide level?
3) If you add the control group data to Figure 1-3 and enter the p-value, you can omit Figure 5-7.
4) Please add the p-value to Table 4.
・Are the conclusions supported by the results?
The text states that there is no correlation between ghrelin levels and sleep disorder parameters. Regression analysis shows a weak correlation between BMI and ghrelin, but was OSA a factor influencing ghrelin levels in multivariate logistic analysis? Could other factors, such as shortening sleep time, affect ghrelin levels?
Comments and Suggestions for Authors
Is it the novelty of this paper that the combination of OSA and severe obesity lowers ghrelin levels?
Author Response
REVIEWER 1
Thank you very much for a valuable review of the article and for your relevant comments.
I'am attaching the text with the corrections as suggested (marked in red font).
Kind regards,
P.Pardak
Recommendations for Authors
Does the introduction provide sufficient background and include all relevant references?
I recommend that you explain a little more detail on the interrelationship between obesity and sleep disorders, and briefly summarize the rest.
The reviewer's suggestions were followed.
Are the methods adequately described?
It is desirable to add to Table 1 about sleep time, sleep test time and number of people according to type3 and type2, comorbidity and medication.
The reviewer's suggestions were followed (Line: 200; Line: 139; Line: 206; respectively)
Are the results clearly presented?
1) Please correct the item of 3.6 because it is duplicated.
The reviewer's suggestions were followed.
2) Although you examined it separately for males and females only in 3.2, was there any difference between males and females in the neuropeptide level?
The reviewer's suggestions were followed. Line: 276.
The levels of neuropeptides were compared between males and females. Males had significantly lower levels of ghrelin than females (Me = 411; range: 212-1928 vs. Me = 499; range: 407-861). However, after including gender in the regression analysis, it turned out to be an insignificant variable. The observed difference probably results from a significantly lower value of BMI and weight in the group of women participating in the study.
No significant differences in the levels of obestatin or leptin depending on gender were found.
Average values |
Gender |
p |
|||||||||||
Males |
Females |
||||||||||||
N |
Me |
s |
min |
max |
N |
Me |
s |
min |
max |
||||
ghrelin [pg/ml] |
48 |
464,8 |
411,0 |
280,1 |
211,5 |
1928,0 |
10 |
554,8 |
499,2 |
170,6 |
406,6 |
861,5 |
0,0372* |
leptin [ng/ml] |
32 |
28,8 |
26,2 |
7,3 |
16,5 |
44,8 |
9 |
29,6 |
28,9 |
6,4 |
21,5 |
41,7 |
0,7218 |
obestatin [pg/ml] |
32 |
64,6 |
61,9 |
9,7 |
53,9 |
98,2 |
9 |
61,8 |
61,6 |
4,2 |
57,0 |
70,3 |
0,5876 |
Performed with Mann-Whitney test
4) Please add the p-value to Table 4.
The reviewer's suggestions were followed.
3) If you add the control group data to Figure 1-3 and enter the p-value, you can omit Figure 5-7.
It would certainly increase the consistency of the article.
However, we were concerned that the superimposition of the three plots (for the whole group, for the sleep apnea group and the control group) would show less clearly the differences in neuropeptide levels depending on the diagnosis of sleep apnea.
Naturally, if this change is recommended the figures will be changed.
Are the conclusions supported by the results?
The text states that there is no correlation between ghrelin levels and sleep disorder parameters. Regression analysis shows a weak correlation between BMI and ghrelin, but was OSA a factor influencing ghrelin levels in multivariate logistic analysis? Could other factors, such as shortening sleep time, affect ghrelin levels?
An additional multivariate analysis was performed, the diagnosis of OSA and sleep time was taken into account as a factor potentially influencing the concentration of ghrelin. As a result of this analysis, no relationship was found between ghrelin levels and diagnosis of OSA, gender and sleep time. Line:320.
Independent variable |
Logarithm of the mean ghrelin concentration R2 = 12,8% F = 2,6 p = 0,0589 |
||
B (95% c.i.) |
p |
ß |
|
BMI |
-0,018 (-0,035; -0,001) |
0,0433* |
-0,27 |
Diagnosis of OSA |
-0,084 (-0,306; 0,137) |
0,4477 |
-0,10 |
Sleep time (min.) |
0,001 (-0,001; 0,002) |
0,3135 |
0,13 |
Comments and Suggestions for Authors
Is it the novelty of this paper that the combination of OSA and severe obesity lowers ghrelin levels?
The above statement can be considered a novelty of this work.
Moreover, a novelty is the confirmation of the significant influence of sleep apnea on the reduction of ghrelin values and the aggravation of obesity-related disturbances in the level of this neuropeptide.

Reviewer 2 Report
The Authors should define the type of the study (observational? non random?).
Overall tables should be improve with vertical lines among studied groups, especially table 1.
The decimal digits in the p values sound redundant, the same in the text.
Figures 5-7 p value shows comma instead of full stop.
Author Response
REVIEWER 2
Thank you very much for a valuable review of the article and for your relevant comments.
I'am attaching the text with the corrections as suggested (marked in blue font).
Kind regards,
P.Pardak
Comments and Suggestions for Authors
The Authors should define the type of the study (observational? non random?).
The research described in the article is an observational study.
I have placed a relevant notes above the title of the manuscript and in the text (L: 98)
Overall tables should be improve with vertical lines among studied groups, especially table 1.
The reviewer's suggestions were followed.
The decimal digits in the p values sound redundant, the same in the text.
Thank you kindly for this remark.
Unfortunately, I have difficulty fully complying with this recommendation, as the second reviewer asked to include the p-value in one of the tables. Therefore, it can be assumed that deleting all p-values will conflict with future recommended corrections.
Therefore, in most places the p-values were removed, they were only left in the places presenting the most important results (their length was also reduced).
Naturally, if a change is necessary, the p-values will be completely removed.
Figures 5-7 p value shows comma instead of full stop.
The reviewer's suggestions were followed.